# Isavuconazole Treatment in a Mixed Patient Cohort with Invasive Fungal Infections: Outcome, Tolerability and Clinical Implications of Isavuconazole Plasma Concentrations

**DOI:** 10.3390/jof6020090

**Published:** 2020-06-22

**Authors:** Christoph Zurl, Maximilian Waller, Franz Schwameis, Tina Muhr, Norbert Bauer, Ines Zollner-Schwetz, Thomas Valentin, Andreas Meinitzer, Elisabeth Ullrich, Stefanie Wunsch, Martin Hoenigl, Yvonne Grinschgl, Juergen Prattes, Abderrahim Oulhaj, Robert Krause

**Affiliations:** 1Section of Infectious Diseases and Tropical Medicine, Department of Internal Medicine, Medical University of Graz, 8036 Graz, Austria; christoph.zurl@medunigraz.at (C.Z.); maximilian.waller@stud.medunigraz.at (M.W.); ines.schwetz@medunigraz.at (I.Z.-S.); thomas.valentin@klinikum-graz.at (T.V.); elisabeth.ullrich@medunigraz.at (E.U.); stefanie.wunsch@medunigraz.at (S.W.); mhoenigl@health.ucsd.edu (M.H.); juergen.prattes@medunigraz.at (J.P.); 2Division of General Paediatrics, Department of Paediatrics and Adolescent Medicine, Medical University of Graz, 8036 Graz, Austria; 3BioTechMed-Graz, 8010 Graz, Austria; 4Department for Anaesthesiology and Intensive Care Medicine, Landesklinikum Baden, 2500 Baden, Austria; franz.schwameis@baden.lknoe.at; 5Department of Internal Medicine, Landeskrankenhaus Graz 2, 8020 Graz, Austria; tina.muhr@kages.at; 6Department of Internal Medicine, Landeskrankenhaus Hartberg, 8230 Hartberg, Austria; norbert.bauer@lkh-hartberg.at; 7Clinical Institute of Medical and Chemical Laboratory Diagnostics, Medical University of Graz, 8036 Graz, Austria; andreas.meinitzer@medunigraz.at; 8Institute of Hygiene, Microbiology and Environmental Medicine, Medical University of Graz, 8036 Graz, Austria; 9Division of Infectious Diseases and Global Public Health, Department of Medicine, University of California San Diego, San Diego, CA 92093, USA; 10Department of Anesthesiology and Intensive Care Medicine, Medical University of Graz, 8036 Graz, Austria; yvonne.grinschgl@medunigraz.at; 11Institute of Public Health, College of Medicine and Health Sciences, United Arab Emirates University, P.O. Box 17666, Al Ain, UAE; aoulhaj@uaeu.ac.ae

**Keywords:** isavuconazole, fungal infections, therapeutic drug monitoring, anti-infective agents, antifungal therapy

## Abstract

Isavuconazole (ISA) is a triazole antifungal agent recommended for treatment of invasive aspergillosis or mucormycosis. The objective of this study was to evaluate ISA levels in a real world setting in a mixed patient cohort including patients with non-malignant diseases and extracorporeal treatments, and to correlate findings with efficacy and safety outcomes. We investigated 33 ISA treatment courses in 32 adult patients with hematological and other underlying diseases and assessed the clinical response, side effects and ISA trough plasma concentrations. ISA treatment led to complete and partial response in 87% of patients and was well tolerated. The median ISA plasma concentration was 3.05 µg/mL (range 1.38–9.1, IQR 1.93–4.35) in patients without renal replacement therapy (RRT) or extracorporeal membrane oxygenation (ECMO) and significantly lower in patients with RRT including cases with additional ECMO or Cytosorb^®^ adsorber therapy (0.88 µg/mL, range 0.57–2.44, IQR 0.71–1.21). After exclusion of values obtained from four patients with ECMO or Cytosorb^®^ adsorber the median concentration was 0.91 µg/mL (range 0.75–2.44, IQR 0.90–1.36) in the RRT group. In addition to previous recommendations we propose to monitor ISA trough plasma concentrations in certain circumstances including RRT, other extracorporeal treatments and obesity.

## 1. Introduction

Isavuconazole (ISA) is a triazole antifungal agent recommended for treatment of invasive aspergillosis or mucormycosis [1,2,3]. In the SECURE trial, ISA was non-inferior to voriconazole for primary treatment of suspected invasive mould disease including mostly patients with invasive aspergillosis, but also cases of non-aspergillus mould or mixed mould infections, and was associated with fewer drug related adverse events [4]. Recently, a post hoc analysis of 36 patients from SECURE showed no relationship between drug exposure (expressed as area under the curve) related to minimum inhibitory concentration (MIC) values of cultured fungi and outcome parameters (i.e., mortality, response of treatment) or safety endpoints (e.g., elevation of liver transaminases). The authors concluded that without a clear relationship of ISA drug exposure and outcome or side effects there is no clinical evidence for recommending routine therapeutic drug monitoring (TDM) compared to, e.g., voriconazole or posaconazole [5,6,7]. Another post hoc analysis of the SECURE trial showed that more than 97% of patients had ISA plasma concentrations >1 µg/mL and <7 µg/mL. Again, no concentration-dependent relationship for efficacy or safety was observed [8].

One larger study investigated 283 plasma concentrations from real world ISA use, and found ISA plasma concentrations of >1 µg/mL in >90% of patients, resembling findings from the SECURE trial [9]. However, no clinical data were available, and therefore the study could not assess potential associations between ISA levels and efficacy and safety events [9]. This limitation was overcome by a second recent study showing that prolonged administrations and high serum levels of ISA may be associated with adverse events [10]. However, that study was limited by a relatively small sample size (19 patients), all with underlying malignant diseases (18 hematological and one solid), necessitating validation of these findings in other trials and other patient cohorts. The need for additional data on ISA TDM is also emphasized in recent TDM guidelines, where only preliminary recommendations for TDM were made pending further published data [11,12].

The objective of this study was to evaluate ISA levels in a real world setting in a mixed patient cohort including patients with non-malignant diseases and extracorporeal treatments, and to correlate findings with efficacy and safety outcomes.

## 2. Materials and Methods

This observational national multicenter cohort study prospectively included patients aged 18 years or older undergoing treatment with ISA at four centers in the Southern and Eastern part of Austria, the Medical University of Graz, the Landeskrankenhaus (LKH) Graz 2, LKH Hartberg, and the Klinikum Baden from November 2016 to October 2019.

Based on recommendations of the local drug advisory committees ISA was allowed to be administered in patients with possible/probable/proven invasive aspergillosis using revised EORTC MSG criteria and/or Blot criteria in case of invasive aspergillosis in critical ill patients [13,14]. Patients were eligible for ISA treatment if fulfilling criteria of probable/proven mucormycosis with contraindications to liposomal Amphotericin B (lipAmpB), adverse events under lipAmpB treatment, lack of clinical response, or in other probable/proven invasive fungal infections based on antifungal susceptibility testing favoring Isavuconazole over alternative antifungals. Treatment consisted of 200 mg q8 h for 48 h followed by 200 mg once daily.

Patients’ medical records were reviewed individually by using a standardized data collection template in order to collect demographic information and clinical data, mycological laboratory test results, ISA trough plasma concentrations, as well as ISA formulation, dosing information, termination of ISA treatment as well as other antifungal therapy, clinical response, serious adverse events, and breakthrough infections. Treatment success, failure and death were classified according to EORTC/MSG criteria [15]. Breakthrough infections under ISA were defined according to revised ECMM/MSG criteria [13,16,17]. Possible side effects like QT abnormalities, neutropenia and hepatotoxicity were observed by electrocardiograms (ECG) and laboratory assessment of blood cell count and liver enzymes such as alanine aminotransferase (AST), aspartate aminotransferase (ALT) and alkaline phosphatase (AP), respectively. Other possible adverse events like headache, abdominal pain, nausea or vomiting were documented by daily assessment during hospitalization or at outpatient visits. Indications of ISA treatments differed between individual patients, so efficacy assessment could not be performed at given time points. As data were extracted from real word clinical cases efficacy was assessed throughout the treatment following by an observational time period including radiological assessments as clinically indicated. This observation was up to two months after end of treatment. Safety was assessed concomitantly.

The day of the first administration of ISA was considered as day 0. Samples for determination of ISA trough plasma concentrations were obtained in the morning immediately prior to scheduled ISA infusion or intake as ordered by the treating physician. As no routine TDM algorithm for ISA was available by the current literature [5,6,7], plasma samples for ISA TDM were obtained as ordered by the treating physicians and adjusted to, e.g., scheduled outpatient visits. ISA plasma concentrations were measured with electrospray ionization tandem mass spectrometry on a Voyager TSQ Quantum triple quadrupole instrument equipped with an Ultimate 3000 chromatography system (Thermo Instruments, San Jose, California, USA) as reported recently [18]. The laboratory investigating ISA plasma concentrations participated in international ISA round-robin tests [18].

The study was approved by the local ethics committee, Medical University Graz, Austria (protocol number 29–444 ex 16/17). All statistical analyses were performed using R version 3.5.1. Continuous data (i.e., ISA plasma concentrations) are presented as medians (inter-quartile ranges [IQR]). The data at patient level were summarized by calculating the median (alternatively maximum) of Isavuconazole levels. More specifically, in patients with more than one single measurement, the median and maximum levels of Isavuconazole were calculated. By doing so, each patient is represented by one single summary measurement (median or maximum) and standard statistical tests can be applied. We used the median as the main variable for comparison between groups and the maximum value was used for sensitivity analysis. Measurements of patients assigned to specified groups (i.e., with or without extracorporal treatments) were considered independent. Group comparisons were performed by Wilcoxon-rank-sum test. A *p*-value of <0.05 was considered significant.

## 3. Results

### 3.1. Study Cohort

A total of 33 courses of ISA treatment were administered to 32 patients (23 (72%) male, nine (28%) female; median age 60 years, range 24 to 85 years, IQR 46–69). Study patients had a median body mass index of 24.6 kg/m^2^ (range 18.5 to 39.6, IQR 23.3–28.5). A total of 14 patients (44%) had hematological diseases (13 malignant), whereas the 18 other patients (56%) had mixed underlying diseases (Table 1).

Fourteen of 32 patients (44%) had proven invasive fungal disease, nine (28%) had probable and nine (28%) had possible invasive fungal disease. One patient with two courses of ISA was classified as suffering from probable invasive fungal disease on both occasions. The median duration of ISA administration was 45 days (min 1, max 441 days, IQR 16–106 days). In 12/33 (36%) ISA courses the patients received only the intravenous and in 6/33 (18%) courses only the oral formulation of ISA. In 15/33 (45%) ISA courses patients received the intravenous and oral formulation. The intravenous ISA formulation was used for a median of 13 days (range 1–54 days, IQR 8–21) whereas the tablet was used for a median of 70 days (range 8–414 days, IQR 34–201). In a total of 20/33 courses (61%) the patients received antifungal therapy or prophylaxis prior to ISA, in six of these 20 courses (30%) ISA was used due to failure and in 13/20 courses (65%) ISA was used due to side effects of preceding antifungals. A total of 11/32 patients (34%) were on renal replacement therapy (RRT; continuous hemodialysis, continuous hemofiltration, or a combination of both), while three of these patients additionally had extracorporeal membrane oxygenation (ECMO) and one additionally had Cytosorb^®^ adsorber therapy (CytoSorbents Europe, Berlin, Germany).

### 3.2. Isavuconazole Efficacy and Safety

Complete response occurred in 18/30 (60%) ISA courses, partial response in 8/30 (27%), and stable disease in 1/30 (3%) ISA courses. Assessment was not possible in three cases. Three out of 32 patients (9%) had a fatal outcome during hospitalization attributable to fungal disease. One patient with pulmonal and cerebral aspergillosis died on day 4 and had no ISA plasma concentrations measured. One patient with possible invasive mould disease died on day 10 of ISA treatment and had 1.36 µg/mL ISA plasma concentration on day 3 (without RRT) and 1.69 µg/mL on day 6 while on continuous hemodialysis. The remaining patient received six days of ISA, was switched to lipAmpB plus voriconazole after receiving the antifungal susceptibility testing of a *Fusarium solani* blood stream isolate and died six weeks later (minimal inhibitory concentrations were 2 µg/mL for Amphotericin B, 1.5 µg/mL for voriconazole and >32 µg/mL for ISA). ISA plasma concentrations were 1.51 µg/mL on day 4, 1.32 µg/mL on day 6 of ISA treatment, and 1.36 µg/mL two days after termination of ISA treatment. In 3/33 (9%) ISA courses no classification was possible (one patient with only one dose of ISA, one patient with acute aortic valve avulsion, one patient with secondary sclerosing cholangitis). Overall, a total of 11/32 patients (34%) had a fatal outcome during hospitalization including the three IFI related deaths. No breakthrough infection during ISA administration was observed.

Six of 33 (18%) ISA treatments led to adverse events including one case with an anaphylaxis (including dyspnea and generalized erythema), one with leucopenia (1.52G/L), two with elevated liver enzymes, one with paraesthesia and one with erythema and elevated liver enzymes. All of these patients had concomitant medication, thus the role of ISA in the reported adverse events remains unclear. All patients recovered fully from adverse events.

### 3.3. Isavuconazole TDM

A total of 145 ISA plasma concentrations were measured. Five ISA plasma concentrations were excluded due to erroneous sampling time points (three samples were not trough levels as samples were obtained after ISA intake, two samples were obtained after termination of ISA treatment). Thus, 140 ISA plasma concentrations were analyzed from 29 courses of ISA treatment with a median of three measurements per patient (range 1–18). There was no change in the dosage due to the measured plasma levels. In three out of four patients with missing ISA plasma concentrations ISA was administered for less than 5 days (single dose, 3 days and 4 days, respectively) while it was administered for one month in one patient. Median time of first measurement after initiation was 2.28 days (range 0.74–9.08, IQR 1.51–2.86). The median plasma concentration was 2.35 µg/mL (range 0.66 to 9.1 IQR 1.49–3.71). After exclusion of values during RRT or ECMO the median ISA concentration was 3.05 µg/mL (range 1.38–9.1, IQR 1.93–4.35) (Figure 1). The median level of seven patients receiving RRT was 0.91 µg/mL (range 0.66–2.44, IQR 0.82–1.36). Compared to values without RRT ISA concentration remained at a lower level over treatment period (Figure 2). After exclusion of values from four patients obtained during RRT and additional extracorporeal treatments (ECMO or Cytosorb^®^ adsorber therapy) the median concentration was 0.91 µg/mL (range 0.75–2.44, IQR 0.90–1.36) in the RRT group. Patients without extracorporal treatments also had higher maximum Isavuconazole trough levels compared to the groups with extracorporal treatments (*p* < 0.001 for RRT; and <0.003 for RRT plus ECMO). Two patients (body mass index 23 and 26 kg/m^2^) with influenza and ARDS received ECMO (iLA activve^®^, Novalung, Heilbronn, Germany) for pulmonary support and continuous RRT. One ECMO patient was treated with a standard dose of ISA for invasive aspergillosis since preceding and dose escalated intravenous voriconazole did not reach target levels of >1 µg/mL. On day 12 after initiation of intravenous ISA the plasma concentration was measured and was 1.79 µg/mL. The patient died 2 days later due to a gangrenous bowel. The other ECMO patient had a standard dose of ISA for treatment of probable intraabdominal *Candida parapsilosis* infection after failure of caspofungin. On day 1 and 4 after initiation of intravenous ISA the plasma concentration was 0.74 µg/mL (during loading dose) and 0.57 µg/mL, respectively. ECMO was terminated on day 6. Subsequent ISA plasma concentration during the second ISA treatment course of this particular patient was 2.44 µg/mL while receiving RRT. The patient died 5 weeks later due to secondary sclerosing cholangitis. The third patient was treated with a Cytosorb^®^ cytokine adsorber within a continuous RRT circuit for 4 days and had an ISA plasma concentration of 1.3 µg/mL immediately prior and 0.82 µg/mL on the last day of adsorber treatment. Subsequent ISA plasma concentrations determined 14 and 32 days after the adsorber treatment but during ongoing RRT were 0.62 µg/mL and 0.91 µg/mL. This patient suffered from pulmonary invasive aspergillosis and had a favorable response after 56 days of ISA treatment. The fourth patient developed ARDS after major cardiac surgery and was treated for pulmonary invasive aspergillosis. ISA concentration was measured daily during a period of 18 days. During ECMO the median ISA concentration was 1.7 µg/mL. On day 12 additional RRT was initiated and ISA concentration decreased to 0.8 µg/mL. ECMO was discontinued on day 15 but RRT was still ongoing and ISA concentration remained below 0.9 µg/mL. Boxplots of ISA plasma concentrations with and without extracorporeal treatments are shown in Figure 1. Overall, there was no correlation of ISA plasma concentrations and complete response or body mass index and ISA plasma concentrations.

Among the six of 33 (18%) ISA treated patients with adverse events, four had ISA plasma concentrations measured, which were always below 5.5 µg/mL (in the two patients who developed anaphylaxis and leucopenia, respectively, no ISA plasma concentrations were measured).

## 4. Discussion

In our cohort ISA was used for treatment of invasive fungal diseases in patients with malignant and non-malignant underlying diseases. Overall, ISA was well tolerated compared to previous studies [4,10]. Eighteen percent had adverse events during treatment including one ISA case with suspected immediate allergic reaction. The rate of side effects in our study was much lower compared to the SECURE trial where systematic prospective assessment revealed that 96% of patients experienced mostly mild treatment-emergent adverse events [4]. In routine clinical setting side effects were previously reported in approximately 30% [10]. ISA treatment led to complete and partial response in 87% of our patients, whereas progression of invasive fungal infection occurred in 9% of ISA treated patients leading to death. Comparable to previous studies no threshold for efficacy could be identified in our study, presumably due to the low number and short term treatments of failure patients.

As influenza might result in ARDS requiring extracorporeal membrane oxygenation (ECMO) and was reported to be an independent risk factor for invasive pulmonary aspergillosis [19] the selection of antifungal drugs for treatment of invasive fungal infection in ECMO patients is of utmost importance. Difficulties of voriconazole and micafungin dosing in patients receiving ECMO due to drug sequestration by the ECMO circuit was described previously [20,21]. In our cohort two patients required veno-venous ECMO for treatment of influenza-associated ARDS and received ISA for invasive aspergillosis or probable invasive candidiasis. Currently, there is no literature reporting on ISA levels in such patients. In this study, three patients were on ECMO and continuous RRT in parallel. Whereas one patient had an ISA trough level of 1.70 µg/mL measured on day 12 of ISA treatment, the second had constantly low ISA levels <1 µg/mL (measured on day 1 and 4 during ECMO but increased to 2.44 µg/mL after termination of ECMO on day 6 but ongoing RRT). The third patient had a median ISA concentration of 1.7 µg/mL during ECMO but experienced lower levels after initiation of additional RRT.

These findings are of course limited by low numbers but may highlight the need of ISA TDM in patients with ongoing ECMO, a fact that was also previously reported for other antifungals like voriconazole or caspofungin. Based on ISA concentrations measured in our ECMO patients one might assume that ISA is sequestrated within the ECMO circuit during the first days of treatment leading to low plasma concentrations according to an assumption previously described for voriconazole [20]. As both patients had ISA plasma concentrations >1 µg/mL while RRT was in place, this extracorporeal treatment obviously had no detrimental effect on ISA plasma concentrations in these particular patients. In the remaining group of patients receiving RRT, one patient had ISA plasma concentrations of 0.62 and 0.67 µg/mL while all the other RRT patients had concentrations >1 µg/mL. Previously, ISA levels were significantly lower after Sustained Low-Efficiency Dialysis (SLED) for treatment of renal impairment in critically ill hematology patients suffering from probable invasive aspergillosis (5.73 μg/mL vs. 3.36 μg/mL; *p* < 0.001; 42% reduction rate of ISA concentration) [22]. Our critically ill patients with RRT received continuous hemodialysis and/or continuous hemofiltration and had a median ISA concentration of 1.12 μg/mL, which was significantly lower compared to patients without RRT. As no ISA was detectable in the ultrafiltrate in the above mentioned study adsorption of ISA to the extracorporeal circuit has to be assumed [22].

Cytokine reduction by hemoadsorption aims to attenuate the overwhelming systemic levels of pro-inflammatory and anti-inflammatory mediators released in the early phase of sepsis. One of our ISA treated patient had a Cytosorb^®^ cytokine adsorber four times for 16 h each within a RRT circuit. Whereas no data of this cytokine adsorber are available determination of ISA plasma concentrations in our patient suggest that this particular device might lower ISA plasma concentrations. However, as the patient subsequently showed ISA plasma concentrations of 0.68 µg/mL and 0.91 µg/mL 14 and 32 days after cytokine adsorption the exact effect of the Cytosorb^®^ cytokine adsorber remains unclear.

One of our patients had a body mass index (BMI) of 39.6 kg/m^2^ (140 kg body weight) which was outside the BMI and weight standard deviations of published ISA data [23,24]. Previously, a significant relationship between BMI and clearance was found hence the suitable ISA dosage in very obese patients is unknown [23]. Despite the high BMI of 39.6 kg/m^2^ the patient received the standard dose of ISA for treatment of probable invasive aspergillosis and was closely monitored by ISA plasma concentrations. The obtained ISA plasma concentrations were 1, 2.42, 3.68 and 3.42 µg/mL, respectively. The patient had a complete response.

A post hoc analysis of ISA exposure-response relationship for measures of efficacy and safety in patients with invasive aspergillosis and infections by other filamentous fungi from the SECURE clinical trial showed that no statistically significant relationship between ISA exposure and either efficacy or safety endpoints. The authors concluded that lack of association between exposure and efficacy indicates that the ISA exposures achieved by clinical dosing were appropriate for treating the infecting organisms in the SECURE study and that side effects were not related to increase in exposures [5]. Comparable ISA plasma concentrations in clinical use and study populations receiving ISA have been observed [9]. However, due to lack of patient-level data assessment of outcome and safety data with regard to ISA concentration was not possible from that study [9]. In an analysis of 19 hematology patients no ISA concentration cut off for efficacy could be identified, but 5 µg/mL was proposed as the upper normal range due to mainly gastrointestinal side effects [10]. In our patient cohort no calculation of treatment failure was possible as deceased patients had ISA treatment durations of only 4–10 days. In patients with adverse events ISA plasma concentrations were always below 5.5 µg/mL. Recently, increased dosage up to 300 mg of ISA per day for treatment of cerebral invasive aspergillosis has been suggested based on two cases [25]. As ISA concentrations within the brain abscess and inflamed meninges were close to ISA plasma concentration but were much lower in normal brain tissue, we assume that ISA TDM might help to adequately dose and manage adverse events in central nervous system aspergillosis cases.

## 5. Conclusions

In summary, ISA was used in patients with and without hematological malignancies for treatment of invasive fungal diseases. ISA was efficacious, well tolerated and showed ISA plasma concentrations comparable to previously successfully treated patients even in cases with high body weight (BMI). Plasma concentrations in patients receiving RRT were significantly lower compared to patients without. However, due to low numbers of patients with high body weight and RRT as well as inconsistent or low levels obtained in patients with ECMO or cytokine adsorber we propose to monitor ISA plasma concentrations in such patients to attain previously suggested but still uncertain target levels of ISA >1 µg/mL [5,9,22].

## Figures and Tables

**Figure 1 jof-06-00090-f001:**
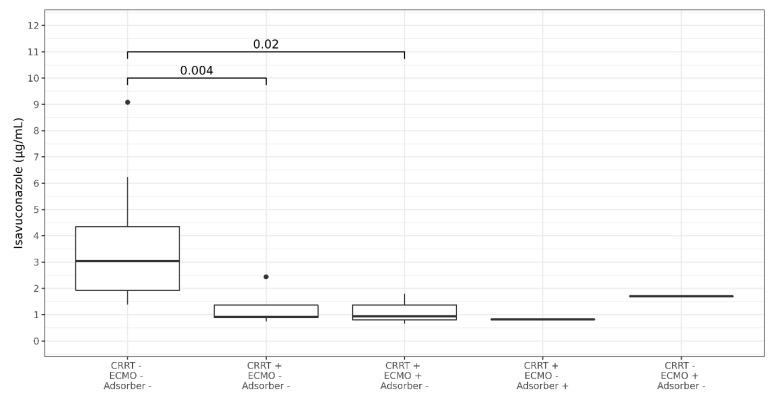
Boxplots showing median Isavuconazole trough plasma concentration in patients with and without extracorporeal treatments. CRRT = renal replacement therapy, ECMO = extracorporeal membrane oxygenation, adsorber = cytosorb adsorber. *P*-Values are shown above the brackets. Dots represent outliers.

**Figure 2 jof-06-00090-f002:**
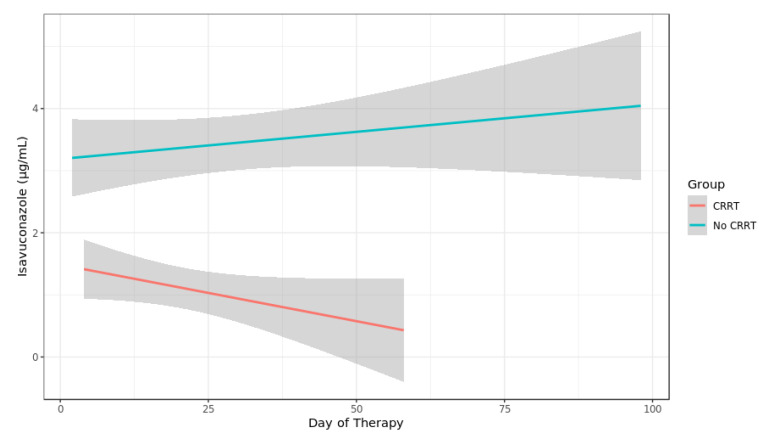
Spaghetti plot of Isavuconazole plasma concentration in patients with and without continuous renal replacement therapy (CRRT).

**Table 1 jof-06-00090-t001:** Demographics of Isavuconazole (ISA) treated patients.

**32 ISA Patients**	
Age, median (IQR), years	60 years (46–69)
Female sex, No. (%)	9 (28%)
Weight, median (IQR), kg	75 (65–84)
Body Mass Index (BMI), median (IQR), kg/m^2^	24.6 (23.3–28.5)
<18.5: underweight, No. (%)	0 (0%)
18.5 to <25: normal, No. (%)	19 (59%)
25 to <30: overweight, No. (%)	7 (22%)
≥30: obese, No. (%)	6 (19%)
**Invasive fungal infection (IFI), EORTC (33 ISA courses *)**	
Possible, No. (%)	9 (27%)
Probable, No. (%)	10 (30%) *
Proven, No. (%)	14 (42%)
**Underlying diseases**	
Hematological disease **	14 (44%)
Solid cancer	2 (6%)
Solid organ transplantation	4 (13%)
Collagenosis/autoimmune diseases	2 (6%)
Type 2 diabetes	2 (6%)
Respiratory tract diseases	3 (9%)
Bacterial infections	3 (9%)
Coronary heart disease	1 (3%)
Trauma associated osteomyelitis	1 (3%)

* One patient with two courses of ISA had probable IFI on both occasions. ** Acute myeloid leukemia 5 patients, aplastic anemia 1 patient, acute lymphatic leukemia 2 patients, chronic lymphatic leukemia 1 patient, lymphoma 4 patients, hemophagocytic syndrome 1 patient.

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
