# Peer review of "Isavuconazole Treatment in a Mixed Patient Cohort with Invasive Fungal Infections: Outcome, Tolerability and Clinical Implications of Isavuconazole Plasma Concentrations"

_jof, 2020, doi:10.3390/jof6020090_

Round 1
Reviewer 1 Report
In this work, the investigators measured isavuconazole trough concentrations using mass spectrometry in 32 adult patients. They found lower concentrations in patients who received standard dose of isavuconazole while being on RRT, ECMO or Cytosorb adsorber therapy. The article addresses an important clinical question. My detailed comments below:
2. Methods. Multiple levels were measured from each patient (median 3, up to 18). These measurements are not independent. The Wilcoxon rank sum test cannot be applied. Please consult with a statistician.
3.1 Study cohort. What was the follow-up time?
3.2 Efficacy and safety. At what day post treatment was response assessed?
Did breakthrough infections occur in the cohort (note on breakthrough infections made under Methods)?
I would be interested to know whether patients with elevated liver enzymes had higher isavuconazole levels.
3.3 Isavuconazole TDM. How soon after initiation of treatment was the first drug level obtained (median, range)?
Why were multiple levels checked? Was the dose ever adjusted based on drug levels?
4. Discussion. I would minimize the use of numerical values in this section in order to make the manuscript easier to read.
The authors may include a brief discussion on whether trough vs random levels should be measured given the long half-life of isavuconazole.
Lines 250-255, "Cytokine reduction … might lower ISA plasma concentrations". Would rephrase the paragraph. It is unknown whether the adsorber lowers isavuconazole levels.
Minor comment: The introduction, methods section and section on isavuconazole TDM can be split into more than one paragraphs.
Author Response
We thank reviewer 1 for thorough review of our manuscript and the helpful suggestions
Reviewer 1
Multiple levels were measured from each patient (median 3, up to 18). These measurements are not independent. The Wilcoxon rank sum test cannot be applied. Please consult with a statistician.
Response
Based on recommendations from our statistical advisor we decided to use statistical methods for independent variables for calculation of differences between the groups as depicted in figure 1, i.e. the patient group without extracorporal treatments compared to patients with different extracorporal treatments. We did not compare ISA levels at different time points but grouped ISA levels based on presence or absence of extracoporal treatments. Isavuconazole levels were measured as indicated and ordered by the treating physician. So, different numbers of ISA levels were obtained in individual patients (range of 1-18 ISA levels as mentioned in the manuscript). The ISA levels were used for calculation of median ISA levels and IQRs in given groups, i.e. without extracorporal treatments, with RRT, with ECMO, with both extracorporal treatments and RRT plus adsorber (only one measurement). Only three patients had measurements that fit in two or more groups and could therefore be considered dependent variables, i.e. ISA levels without extracorporal treatments and later on ISA levels with extracorporal treatments or vice verca. We therefore had a mixture of 29 patients with ISA levels in only one group and 3 patients with ISA levels that fell in 2 or more groups. For calculation of data as depicted in figure 1 ISA levels without extracorporal treatments were omitted for patients with ISA measurements in this and other groups which resulted in comparisons as mentioned in the manuscript, i.e. statistical significant higher levels in patients without extracorporal treatments compared to patients with extracorporal treatments (due to lower numbers of ISA levels in patient groups with extracorporal treatments ISA levels measured during the extracorporal treatment were retained and used for calculations and those ISA levels measured at time points without extracorporal treatments prior or after extracorporal treatment were omitted). However, we described the course of ISA levels in such patients in the main text to provide all of the ISA concentration informations and furthermore included all of the ISA levels in the description of the overall ISA median concentration as mentioned in the manuscript main text. In summary, by using the strategy as described above ISA levels in the group without extracorporal treatment and with extracorporal treatments are independent and Wilcoxon rank sum test is appropriate.
Reviewer 1
Study cohort. What was the follow-up time?
Response
The follow up time was up to two months after treatment as clinically indicated.
Reviewer 1
Efficacy and safety. At what day post treatment was response assessed?
Response
As indications of Isavuconazole treatments differed between individual patients efficacy assessment was not performed at given predefined time points. As data were extracted from real word clinical cases efficacy was assessed throughout the treatment following by an observational time period including radiological assessments as clinically indicated. This observation was up to two months after end of treatment. Safety was assessed concomitantly.
To clarify all of this issues some parts of the material and method section were modified in the revised version of the manuscript to include this information (line 99).
Reviewer 1
Did breakthrough infections occur in the cohort (note on breakthrough infections made under Methods)?
Response
No breakthrough infections were observed during the study. This information was included in the results section of the revised manuscript (line 158).
Reviewer 1
I would be interested to know whether patients with elevated liver enzymes had higher isavuconazole levels.
Response
There were 3 patients with elevated liver enzymes during ISA treatment with one patient having additionally extracorporal treatment. These patients did not have elevated ISA plasma levels.
Reviewer 1
Isavuconazole TDM. How soon after initiation of treatment was the first drug level obtained (median, range)? Why were multiple levels checked? Was the dose ever adjusted based on drug levels?
Response
Median time of first measurement after initiation was 2.28 days (range 0.74 – 9.08m, IQR 1.51 – 2.86). The respective data were added to the revised version of the manuscript (line 177).
Due to missing recommendations of ISA TDM levels were checked for multiple time points as ordered by the treating physician. There was no adjustment of the dosage by the treating physicians depending on the plasma concentration. One patient had a dose reduction due to increased liver enzymes, but this modification of the dosage did not alter ISA plasma concentrations.
Reviewer 1
I would minimize the use of numerical values in this section in order to make the manuscript easier to read.
Response
In our point of view the discussion contains relevant issues of patients with extracorporal treatments regarding ISA concentrations. We believe that the values given in this section are very important for the discussion of the overall data. We therefore kindly ask to maintain the values as they are.
Reviewer 1
The authors may include a brief discussion on whether trough vs random levels should be measured given the long half-life of isavuconazole.
Response
We thank the reviewer for this important thought. We agree with the reviewer that ISA concentrations could eventually be measured as trough levels or randomly due to the long half life of ISA. However, to be in line with TDM of other antifungal agents and therefore to assure correct TDM of all antifungal drugs in clinical settings we suggest to measure trough levels including ISA.
Reviewer 1
Lines 250-255, "Cytokine reduction … might lower ISA plasma concentrations". Would rephrase the paragraph. It is unknown whether the adsorber lowers isavuconazole levels.
Response
The reviewer raised the issue that it is unknown whether the adsorber lowers ISA levels. In the addressed part of the manuscript we discussed as follows: „Whereas no data of this cytokine adsorber is available determination of ISA plasma concentrations in our patient suggest that this particular device might lower ISA plasma concentrations. However, as the patient subsequently showed ISA plasma concentrations of 0.68µg/ml and 0.91µg/ml 14 and 32 days after cytokine adsorption the exact effect of the Cytosorb® cytokine adsorber remains unclear.“ We believe that our phrasing (and especially the last sentence), does not imply that the cytokine adsorber is now known to lower ISA levels. We fully agree with the reviewer that the influence of the cytokine adsorber on ISA levels is not known and therefore ended with the statement that „…the exact effect of the Cytosorb® cytokine adsorber remains unclear.“
Reviewer 1
Minor comment: The introduction, methods section and section on isavuconazole TDM can be split into more than one paragraphs.
Response
The revised manuscript was modified according to the suggestion of the reviewer.
Reviewer 2 Report
The authors describe real life isavuconazole plasma concentrations and tolerability in several populations of patients. The manuscritp is interesting for the readers of the journal, as the authors find lower isavuconazole through levels in patients renal replacement therapy and extracorporeal membrane oxygenation. In general, the manuscript is well written, with appropiate use of tables and figures. Some paragraphs can be reorder in this reviewers opinion. (see below)
Major suggestion
1 In the introduction the authors mention that the different studies did not find a correlation between serum concentrations of isavuconazole and outcome/side effects. Furthermore the mention the objective of this study was to correlate the concentrations found in their study with efficacy. Similar to earlier studies, the authors failed to find an correlation. Possibly (or likely) due to the low number of MIC tests. In this introduction, the main findings of the study were not mentioned of introduced. Furthermore, the authors start the discussion with the fact they couldnt find a correlation with efficacy. I believe more emphasis can be placed on the most important findings (low ISA concentrations in some populations) of the study in the introduction bus also in the discussion.
2 it is not clear in the methods section whether ISA dosages were adjusted based on plasma concentrations.
3 the patients with low voriconazole serum concentrations that changed to ISA maybe a bias. The low concentrations of ISA maybe due to the ECMO, but may also be due to an independent risk factor unrelated to ECMO. As only few patients had ECMO the impact of such bias maybe relatively high, although the effect would still be significant if this patient was excluded.
Minor suggestions
1 Abstract: abbreviate Isavuconazole in line 35. 36
2 species names should be in italics
Author Response
We thank reviewer 2 for thorough review of our manuscript and the helpful suggestions
Reviewer 2
The authors describe real life isavuconazole plasma concentrations and tolerability in several populations of patients. The manuscript is interesting for the readers of the journal, as the authors find lower isavuconazole through levels in patients renal replacement therapy and extracorporeal membrane oxygenation. In general, the manuscript is well written, with appropiate use of tables and figures.
Response
We thank the reviewer for this assessment.
Reviewer 2
In the introduction the authors mention that the different studies did not find a correlation between serum concentrations of isavuconazole and outcome/side effects. Furthermore they mention the objective of this study was to correlate the concentrations found in their study with efficacy. Similar to earlier studies, the authors failed to find an correlation. Possibly (or likely) due to the low number of MIC tests. In this introduction, the main findings of the study were not mentioned of introduced. Furthermore, the authors start the discussion with the fact they couldnt find a correlation with efficacy. I believe more emphasis can be placed on the most important findings (low ISA concentrations in some populations) of the study in the introduction bus also in the discussion.
Response
We thank the reviewer for his/her comment and suggestions. We agree with the reviewer that one major issue in correlation of certain ISA concentrations and efficacy might be the performance of additional MIC measurements and subsequent correlation of ISA concentrations with treatment outcome of patients infected by given microorganisms and their MICs. In addition, some other factors have to be considered like location of diseases (lungs, CNS, others), extent of disease, co-medications and many others. As suggested by the reviewer we start the discussion with the main findings, e.g. the tolerability of ISA and the overall efficacy, followed by the missing threshold for efficacy.
Reviewer 2
It is not clear in the methods section whether ISA dosages were adjusted based on plasma concentrations.
Response
ISA dosage consisted of 200 mg q8h for 48 hours followed by 200 mg once daily. As outlined in the response to reviewer 1 there were no adjustments of the dosage by the treating physician depending on the plasma concentration.
Reviewer 2
The patients with low voriconazole serum concentrations that changed to ISA maybe a bias. The low concentrations of ISA maybe due to the ECMO, but may also be due to an independent risk factor unrelated to ECMO. As only few patients had ECMO the impact of such bias maybe relatively high, although the effect would still be significant if this patient was excluded.
Response
We agree with the reviewer that many other factors might influence voriconazole serum concentrations although the effect of ECMO on voriconazole concentrations has been described in previous literature. However, we did not describe ECMO as the sole reason for low voriconazole concentration but just mention the clinical fact that the patient was on voriconazole with ECMO in place and was switched to ISA due to low voriconazole concentrations. We therefore can not follow the suggestion of the reviewer to exclude this particular patient.
Reviewer 2, minor suggestions
Abstract: abbreviate Isavuconazole in line 35. 36
Response
The abstract was changed accordingly within the revised version of the manuscript.
Reviewer 2, minor suggestion
Species names should be in italics.
Response
The text was adapted according to the suggestion within the revised version of the manuscript.
Round 2
Reviewer 1 Report
The authors did not address the main comment: Multiple measurements on the same patient are not independent (whether the patient is on ECMO or not). I suggest that a statistical advisor weighs in.
Author Response
Reviewer 1
The authors did not address the main comment: Multiple measurements on the same patient are not independent (whether the patient is on ECMO or not). I suggest that a statistical advisor weighs in.
Response
As recommended we contacted an external statistical advisor (Dr. Abderrahim Oulhaj, Associate Professor in Biostatistics, Institute of Public Health, College of Medicine and Health Sciences, United Arab Emirates University) and describe the statistical procedures as follows:
We agree with the reviewer that one of the main assumptions of the Wilcoxon rank sum test is that observations should be independent. To resolve this issue, we summarized the data at patient level by calculating the median (alternatively the maximum) of Isavuconazole levels. More specifically, in patients with more than one single measurement, the median and maximum levels of Isavuconazole were calculated. By doing so, each patient is now represented by one single summary measurement (median or maximum) and the Wilcoxon rank sum test can be applied to each of these variables. In this paper, we used the median as the main variable for comparison between groups and the maximum value was used for sensitivity analysis.
As recommended by the reviewer and to improve the comprehensibility of the statistical procedures we modified this part of the manuscript (line 118) to:
“The data at patient level was summarized by calculating the median (alternatively maximum) of Isavuconazole levels. More specifically, in patients with more than one single measurement, the median and maximum levels of Isavuconazole were calculated. By doing so, each patient is represented by one single summary measurement (median or maximum) and standard statistical tests can be applied. We used the median as the main variable for comparison between groups and the maximum value was used for sensitivity analysis. Group comparisons were performed by Wilcoxon-rank-sum test. A p-value of <0.05 was considered significant.“
According to this strategy the absolute values had to be adapted in the results section and within the abstract to fit with the values as depicted in figure 1.
In the patient with renal replacement therapy and cytosorb adsorber we had only one Isavuconazole measurement during this extracorporal procedure and comparison was therefore not performed for this particular measurement. There was one patient with ECMO but without RRT (and without adsorber) providing 11 Isavuconazole values. As this patient represented the only one in that particular extracorporal treatment group comparison of the median Isavuconazole level and maximum Isavuconazole level of this patient with levels from patients without extracorporal treatments was not performed.
Due to the recommended involvement of additional statistical expertise we kindly ask to include the additional statistical advisor Dr. Abderrahim Oulhaj (Associate Professor in Biostatistics, Institute of Public Health, College of Medicine and Health Sciences, United Arab Emirates University) within the group of authors. The additional statistical advisor reevaluated e.g. the design of the project, the assignment of patients within certain groups, the assignment of Isavuconazole levels within the groups and the statistical strategy and calculation of differences between the groups as well as presentation of data. The inclusion of the statistical advisor has been approved by all of the authors.
